# Healthy lifestyle, endoscopic screening, and colorectal cancer incidence and mortality in the United States: A nationwide cohort study

Kai Wang[1], Wenjie Ma[2,3], Kana Wu[1,4,5], Shuji Ogino[1,6,7,8], Andrew T. Chan[2,3,5,8,9], Edward L. Giovannucci[1,4,5], Mingyang Song[1,2,3,4]*

**1** Department of Epidemiology, Harvard T.H. Chan School of Public Health, Boston, MA, United States of America, **2** Clinical and Translational Epidemiology Unit, Massachusetts General Hospital and Harvard Medical School, Boston, MA, United States of America, **3** Division of Gastroenterology, Massachusetts General Hospital and Harvard Medical School, Boston, MA, United States of America, **4** Department of Nutrition, Harvard T.H. Chan School of Public Health, Boston, MA, United States of America, **5** Channing Division of Network Medicine, Department of Medicine, Brigham and Women's Hospital and Harvard Medical School, Boston, MA, United States of America, **6** Department of Oncologic Pathology, Dana-Farber Cancer Institute and Harvard Medical School, Boston, MA, United States of America, **7** Program in MPE Molecular Pathological Epidemiology, Department of Pathology, Brigham and Women's Hospital and Harvard Medical School, Boston, MA, United States of America, **8** Broad Institute of MIT and Harvard, Cambridge, MA, United States of America, **9** Department of Immunology and Infectious Diseases, Harvard T.H. Chan School of Public Health, Boston, MA, United States of America

* mis911@mail.harvard.edu

**Data Availability Statement:** Because of the sensitive nature of the data collected for this study, requests to access the dataset from qualified

## Abstract

### Background

Healthy lifestyle and screening represent 2 major approaches to colorectal cancer (CRC) prevention. It remains unknown whether the CRC-preventive benefit of healthy lifestyle differs by endoscopic screening status, and how the combination of healthy lifestyle with endoscopic screening can improve CRC prevention.

### Methods and findings

We assessed lifestyle and endoscopic screening biennially among 75,873 women (Nurses' Health Study, 1988 to 2014) and 42,875 men (Health Professionals Follow-up Study, 1988 to 2014). We defined a healthy lifestyle score based on body mass index, smoking, physical activity, alcohol consumption, and diet. We calculated hazard ratios (HRs) and population-attributable risks (PARs) for CRC incidence and mortality in relation to healthy lifestyle score according to endoscopic screening. Participants' mean age (standard deviation) at baseline was 54 (8) years. During a median of 26 years (2,827,088 person-years) follow-up, we documented 2,836 incident CRC cases and 1,013 CRC deaths. We found a similar association between healthy lifestyle score and lower CRC incidence among individuals with and without endoscopic screening, with the multivariable HR per one-unit increment of 0.85 (95% CI, 0.80 to 0.90) and 0.85 (95% CI, 0.81 to 0.88), respectively (P-interaction = 0.99). The fraction of CRC cases that might be prevented (PAR) by endoscopic screening alone was 32% (95% CI, 31% to 33%) and increased to 61% (95% CI, 42% to 75%) when combined with healthy lifestyle (score = 5). The corresponding PAR (95% CI) increased from 15% (13% to

researchers trained in human subject confidentiality protocols may be sent to Brigham and Women's/Harvard Cohorts at https://docs.google.com/forms/d/e/1FAIpQLScAPV23ZIBpkk9CyEJ1OcFJjMol9eIKEpLYnPu7g3PgBL57XA/viewform.

**Funding:** This work was supported by the American Cancer Society Mentored Research Scholar Grant (MRSG-17-220-01 - NEC to M.S.); by the U.S. National Institutes of Health (NIH) grants [P01 CA87969 to M.J. Stampfer; U01 CA186107 to M.J. Stampfer; P01 CA55075, to W. C. Willett; UM1 CA167552 to W.C. Willett; U01 CA167552 to L.A. Mucci and W.C. Willett; K24 DK098311, R01 CA137178, R01 CA202704, R01 CA176726, to A.T.C.; K99 CA215314 and R00 CA215314 to M.S.; R35 CA197735 and R01 CA151993 to S.O.]. Dr. Chan is a Stuart and Suzanne Steele MGH Research Scholar. The funders had no role in design and conduct of the study; collection, management, analysis and interpretation of the data; preparation, review, or approval of the manuscript; and decision to submit the manuscript for publication.

**Competing interests:** I have read the journal's policy and the authors of this manuscript have the following competing interests: Andrew T. Chan previously served as a consultant for Bayer Pharma AG, Pfizer Inc., and Janssen Pharmaceuticals for work unrelated to the topic of this manuscript. This study was not funded by Bayer Pharma AG, Pfizer Inc., or Janssen Pharmaceuticals. No other conflict of interest exists.

**Abbreviations:** AJCC, American Joint Committee on Cancer; BMI, body mass index; CRC, colorectal cancer; HPFS, Health Professionals Follow-up Study; HR, hazard ratio; NHS, Nurses' Health Study; PAR, population-attributable risk; RR, relative risk; STROBE, Strengthening the Reporting of Observational Studies in Epidemiology; TNM, tumor–node–metastasis; WCRF/AICR, World Cancer Research Fund/American Institute for Cancer Research.

16%) to 51% (17% to 74%) for proximal colon cancer and from 47% (45% to 50%) to 75% (61% to 84%) for distal CRC. Results were similar for CRC mortality. A limitation of our study is that our study participants are all health professionals and predominantly whites, which may not be representative of the general population.

## Conclusions

Our study suggests that healthy lifestyle is associated with lower CRC incidence and mortality independent of endoscopic screening. An integration of healthy lifestyle with endoscopic screening may substantially enhance prevention for CRC, particularly for proximal colon cancer, compared to endoscopic screening alone.

## Author summary

### Why was this study done?

- Colorectal cancer is the third leading cause of cancer death in both men and women in the United States.

- Healthy lifestyle and screening represent 2 major approaches to colorectal cancer prevention.

- Endoscopic screening can detect and remove precancerous lesions, but it remains unknown whether individuals having undergone endoscopic screening can still benefit from healthy lifestyle.

- It remains to be determined to what extent a combination of healthy lifestyle and endoscopic screening can improve colorectal cancer prevention compared with endoscopic screening alone.

### What did the researchers do and find?

- Using 2 large nationwide cohorts in the US with a median of 26 years of follow-up, we prospectively examined the association of healthy lifestyle with colorectal cancer incidence and mortality among individuals having and not having undergone endoscopic screening separately.

- We also estimated the proportion of colorectal cancer cases and deaths that are potentially prevented by endoscopic screening alone and by a combination of healthy lifestyle and endoscopic screening.

- We found a similar association between healthy lifestyle and lower incidence and mortality of colorectal cancer among individuals having and not having undergone endoscopic screening.

- We also found that an integration of healthy lifestyle with endoscopic screening might substantially enhance the prevention of colorectal cancer and related death compared with endoscopic screening alone, and the increment of the beneficial association was particularly high for proximal colon cancer.

**What do these findings mean?**

- Our results indicate that healthy lifestyle is associated with lower colorectal cancer incidence and mortality independent of endoscopic screening status.

- The evidence highlights the potential of healthy lifestyle to complement endoscopic screening for optimal prevention of colorectal cancer.

- Our findings highlight the particular merit of an integration of healthy lifestyle and endoscopic screening for prevention of proximal colon cancer, for which endoscopic screening has limited effectiveness.

- These findings suggest the value of healthy lifestyle for colorectal cancer prevention in settings lacking coverage or capacity of endoscopic screening.

## Introduction

Colorectal cancer (CRC) is the third most commonly diagnosed cancer (40.7 per 100,000 population) and the third leading cause of cancer death (14.8 per 100,000 population) in the United States (US) according to the estimates in 2020 [1]. Screening has substantially contributed to the decline in CRC incidence and mortality in elderly Americans during the past 2 decades through detection and removal of precancerous lesions [2–5]. According to the most recent National Health Interview Survey in 2015, the prevalence of recommendation-consistent CRC screening has increased from 43% in 2005 to 63% in 2015 among US adults aged 50 years and older [6].

Healthy lifestyle practices, including no or quitted smoking, maintaining a healthy body weight, being physically active, limiting alcohol intake, and eating a healthy diet, have been associated with a substantial reduction in CRC incidence and mortality [7–12]. Compelling data indicate that 20% to 70% of CRC cases and deaths could potentially be prevented by adhering to a healthy lifestyle [10–12], highlighting the essential role of healthy lifestyle in CRC prevention. However, adoption and long-term maintenance of healthy lifestyle are challenging, particularly due to the reported "health certificate effect" of CRC screening, in which individuals with negative endoscopic screening may perceive themselves as certified healthy and have reduced incentives for the adoption of healthy lifestyle [13–15]. Nevertheless, it is unknown whether healthy lifestyle in individuals having already undergone endoscopic screening remains as beneficial for CRC prevention as those without endoscopic screening. Furthermore, how an integration of healthy lifestyle with endoscopic screening can improve CRC prevention compared with endoscopic screening alone is also an open question. Given the rising use of endoscopic screening, addressing these questions has important clinical and public health significance for improved CRC prevention.

Therefore, in the current study, we assessed the association of healthy lifestyle with CRC incidence and mortality among individuals having and not having undergone endoscopic screening within 2 large cohorts in the US, including the Nurses' Health Study (NHS) and Health Professionals Follow-up Study (HPFS). We also calculated the population-attributable risks (PARs) for CRC incidence and mortality associated with endoscopic screening alone and a combination of healthy lifestyle and endoscopic screening.

## Methods

### Study population

The NHS included 121,700 US female nurses aged 30 to 55 years at enrollment in 1976; and the HPFS included 51,529 US male health professionals aged 40 to 75 years at enrollment in 1986 [16,17]. In both cohorts, participants completed a detailed questionnaire about their lifestyle and medical information at baseline, and every 2 years thereafter, with over 90% of follow-up [18]. In the current study, we defined baseline as the year of 1988 for both NHS and HPFS, when we started to collect information of endoscopy. Written informed consent was obtained from all participants. We conducted this study following a prospective analysis proposal (S1 Text). The study protocol was approved by the institutional review board of the Brigham and Women's Hospital and Harvard T.H. Chan School of Public Health, and those of participating registries as required. This study was reported as per the Strengthening the Reporting of Observational Studies in Epidemiology (STROBE) guideline (S1 Checklist).

Among participants who returned the baseline questionnaires (1988 for both cohorts), we excluded those who had a history of cancer (except nonmelanoma skin cancer), or ulcerative colitis; with a body mass index (BMI) of $<18.5$ kg/m$^2$; reported implausible energy intakes ($<500$ or $>3,500$ kcal/d for women; $<800$ or $>4,200$ kcal/d for men); or with missing data on lifestyle exposures or endoscopic screening. After these exclusions, 75,873 women and 42,875 men were included in the analysis (see flowchart in S1 Fig).

### Assessment of lifestyle factors

We considered 5 lifestyle factors: BMI, cigarette smoking, physical activity, diet, and alcohol intake. Height, body weight, cigarette smoking, and physical activity were self-reported through biennial questionnaires. Physical activity was assessed by the total hours per week for moderate-to-vigorous intensity activity (including brisk walking) that requires the expenditure of at least 3 metabolic equivalents per hour. Alcohol use and diet were self-reported every 4 years using validated food frequency questionnaires [19,20]. Diet quality was assessed using the 6 dietary recommendations by the World Cancer Research Fund/American Institute for Cancer Research (WCRF/AICR) Third Expert Report released in 2018 for red meat, processed meat, dietary fiber, dairy products, whole grains, and calcium supplement use [21]. To capture long-term exposures and reduce random within-person variation, we calculated cumulative average levels of the exposures. When data on lifestyle factors were missing in a given questionnaire cycle, the last nonmissing observation was carried forward. Detailed information of lifestyle and covariates assessment is provided in S2 Text.

For each of the 5 lifestyle factors, we defined a binary criterion, by which the participants received a score of 1 if they met the criterion and 0 otherwise. The healthy lifestyle comprised a BMI of $\geq 18.5$ and $<25.0$ kg/m$^2$, never smoking or past smoking with pack-years $<5$, moderate-to-vigorous intensity activity for $\geq 30$ minutes per day, none-to-moderate alcohol intake ($<1$ drink [14 g alcohol] per day for women and $<2$ drinks per day for men), and meeting at least 3 of the 6 dietary recommendations by the WCRF/AICR Third Expert Report 2018. An overall healthy lifestyle score (range, 0 to 5) was then calculated by summing the 5 scores, with a higher score indicating a healthier lifestyle.

### Assessment of endoscopic screening

In both cohorts, beginning in 1988 and continuing through 2002, participants were asked biennially whether they had undergone lower gastrointestinal endoscopy and, if so, the reason for the endoscopy. In 2004, we additionally inquired whether the previously reported

endoscopies were sigmoidoscopies or colonoscopies. In every cycle thereafter, responses for sigmoidoscopy and colonoscopy were recorded separately. In random samples of participants who reported having negative endoscopy (n = 114 in the NHS and 140 in the HPFS), we collected endoscopic records and observed high concordance rate with self-reported negative endoscopy (97% in the NHS and 100% in the HPFS) [22–24].

Participants were considered as endoscopically unscreened until the first time they reported undertaking endoscopy for screening purpose and as endoscopically screened thereafter for the remainder of follow-up. Given our focus on screening endoscopy, we stopped updating endoscopy information in mortality analysis once a participant is diagnosed with CRC.

## Ascertainment of cases and deaths of CRC

Participants reported CRC diagnoses on each biennial questionnaire. CRC deaths were identified through the National Death Index or reported by family members [25]. For the reported CRC cases or deaths, we obtained medical records to confirm the diagnosis or cause of death by study physicians. When we were unable to obtain medical records (approximately 10% of cases), we linked to the appropriate cancer registry to confirm the diagnosis or death.

We classified CRCs into proximal colon cancers that encompassed those occurring in the cecum, ascending colon, hepatic flexure, and transverse colon; and distal CRCs that encompassed those occurring in the splenic flexure, descending colon, sigmoid colon, rectosigmoid junction, and rectum. We also staged CRC according to the American Joint Committee on Cancer (AJCC) tumor–node–metastasis (TNM) cancer staging system [26].

## Statistical analysis

Details about statistical analysis are provided in S2 Text. Participants contributed person-time from return of the baseline questionnaire (1988 for both cohorts) until the date of CRC diagnosis (for CRC incidence analysis only), death, loss to follow-up, or end of the follow-up period (June 30, 2014 for the NHS and January 31, 2014 for the HPFS), whichever came first. We first calculated the age-adjusted CRC incidence and mortality rates by healthy lifestyle score and endoscopic screening status. To account for multiple records per participant and time-varying exposures and covariates, we used an Andersen-Gill data structure with a new record for each 2-year follow-up period. Then, we used time-varying and age-, period-, and cohort-stratified Cox proportional hazards regression models to estimate the multivariable hazard ratios (HRs) and 95% confidence intervals (CIs) for the associations of individual and combined lifestyle factors with CRC incidence and mortality according to endoscopic screening status. We also assessed the associations by anatomical site (proximal colon cancer or distal CRC) and AJCC TNM stage of CRC (TNM 1 and 2 or 3 and 4) using the subtype-stratified Cox proportional cause-specific hazards regression model by the duplication method [27].

To examine the benefit of an integration of healthy lifestyle with endoscopic screening compared to endoscopic screening alone, we calculated the hypothetical PARs [28] and 95% CIs [10] to estimate the proportion of CRC cases and deaths that could have been avoided if all participants in the cohorts had undertaken endoscopic screening and/or maintained healthy lifestyle. To calculate the PARs, we derived the exposure prevalence within our cohorts and used multivariable pooled logistic regression models to calculate the relative risk (RR) [29]. Using the prevalence ($P_i$) at the exposure category i and the corresponding RR estimates ($RR_i$), we calculated the PAR for healthy lifestyle using the following approximate formula: $PAR =$

$\frac{\sum P_i(HR_i-1)}{\sum P_i(HR_i-1)+1}$ [30]. To examine the incremental benefit, we ran the PAR analysis for different definitions of healthy lifestyle (score of $\geq 2$, $\geq 3$, $\geq 4$, or = 5).

In secondary analyses, we calculated the PARs according to family history of CRC (yes and no), regular aspirin use (yes and no), age ($\leq 65$, 66–75, and >75 years), and sex (female and male). Because CRC screening is generally recommended for individuals aged 50 to 75 years [6,31], we repeated our analyses among participants aged 50 to 75 years only. To examine whether the results were driven differently by colonoscopic and sigmoidoscopic screenings, we repeated the analyses by only focusing on colonoscopic screening while excluding the person-time counted in the sigmoidoscopic screening group. Due to that we carried forward the status of endoscopic screening since the first time they reported taking endoscopic screening, as a sensitivity analysis, we suspended updating the lifestyle information since the first endoscopic screening and repeated the analysis. In the main analysis, we counted the cycles when no response to endoscopy was recorded (accounting for 11% of all person-time) into the non-screening group, so we repeated the analysis after excluding the cycles as a sensitivity analysis. Because the binary variables could not account for the gradient in CRC risk with more extreme levels of these lifestyle factors, we conducted another sensitivity analysis by calculating an expanded healthy lifestyle score using more refined categorizations for the lifestyle factors. We assigned scores of 1 (least healthy) to 5 (most healthy) to the categories of the lifestyle factors and summed the scores across all 5 factors (range, 5 to 25). We then categorized this expanded score into 6 levels (5 to 10, 11 to 13, 14 to 16, 17 to 19, 20 to 22, and 23 to 25), which were defined as the new healthy lifestyle score (0, 1, 2, 3, 4, and 5). Finally, to examine the benefit of healthy lifestyle for overall health, we assessed mortality due to other causes than CRC. In this analysis, we additionally adjusted for physical exam for disease screening, mammography for breast cancer screening (women only), and prostate-specific antigen testing for prostate cancer screening (men only).

## Results

In the 2 cohorts of 118,748 participants with a median of 26 years (2,827,088 person-years) of follow-up, we documented 2,836 incident CRC cases and 1,013 CRC deaths, among which 1,131 cases (40%) and 375 deaths (37%) were of proximal colon cancer, 1,305 cases (46%) and 433 deaths (43%) were of distal colorectal cancer, with the remaining (14% cases and 20% deaths) having no confirmed site information. Overall, 41% and 59% of person-years occurred in the endoscopic screening and non-screening groups, respectively. As shown in Table 1, compared with the non-screening group, the screening group were older, more likely to be male, have family history of CRC, use aspirin, use multivitamins, and have a healthier lifestyle. Within each of the 2 groups, participants with a healthier lifestyle were more likely to be younger, have family history of CRC, and use multivitamins.

Fig 1 shows the associations of healthy lifestyle score with CRC incidence and mortality according to endoscopic screening status. Each unit increment in the healthy lifestyle score was associated with an HR for CRC incidence of 0.85 (95% CI, 0.80 to 0.90) in the screening group and 0.85 (95% CI, 0.81 to 0.88) in the non-screening group (P-interaction = 0.99) (Fig 1A). Compared with participants without endoscopic screening and living a less healthy lifestyle (score = 0), those with screening and a less healthy lifestyle had an HR of CRC incidence of 0.37 (95% CI, 0.24 to 0.56), those without screening but living the healthiest lifestyle (score = 5) had an HR of 0.38 (95% CI, 0.26 to 0.56), those with both screening and a healthiest lifestyle had an HR of 0.14 (95% CI, 0.09 to 0.23). Likewise, the associations of healthy lifestyle score with CRC mortality were also similar among individuals with and without endoscopic

**Table 1. Age- and sex-standardized characteristics\* of study participants according to endoscopic screening status and healthy lifestyle score†.**

| | Nonendoscopic screening (1,661,951 person-years, 59%) | | | Endoscopic screening (1,165,137 person-years, 41%) | | |
|---|---|---|---|---|---|---|
| | Healthy lifestyle score | | | Healthy lifestyle score | | |
| | 0 | 2 | 4 | 0 | 2 | 4 |
| Person-years (% within group) | 40,813 (2) | 603,843 (36) | 195,339 (12) | 25,379 (2) | 383,484 (33) | 180,278 (15) |
| Age, years | 62.7 | 61.9 | 61.2 | 69.0 | 68.7 | 68.7 |
| Male sex, % | 38 | 29 | 38 | 44 | 39 | 46 |
| White, % | 98 | 96 | 95 | 98 | 95 | 94 |
| Family history of colorectal cancer, % | 12 | 13 | 13 | 19 | 20 | 21 |
| Regular aspirin use, % | 29 | 26 | 25 | 35 | 33 | 32 |
| Current multivitamins use, % | 37 | 39 | 48 | 50 | 51 | 59 |
| Body mass index, kg/m$^2$ | 28.3 | 27.0 | 23.1 | 28.2 | 27.0 | 23.3 |
| Current smoker, % | 30 | 13 | 2 | 13 | 6 | 1 |
| Pack-years of smoking‡ | 34.8 | 24.9 | 12.1 | 32.4 | 23.1 | 11.5 |
| Alcohol intake, g/d | 32.3 | 6.3 | 4.4 | 31.8 | 7.7 | 5.5 |
| Physical activity, min/d | 9.5 | 15.1 | 39.2 | 10.7 | 17.7 | 39.7 |
| Dietary intake | | | | | | |
| Red meat, serving/d | 0.7 | 0.7 | 0.5 | 0.7 | 0.6 | 0.4 |
| Processed meat, serving/d | 0.4 | 0.4 | 0.2 | 0.4 | 0.3 | 0.2 |
| Dietary fiber, g/d | 14.8 | 17.5 | 21.0 | 16.4 | 19.3 | 22.9 |
| Dairy products, serving/d | 2.2 | 2.3 | 2.5 | 2.2 | 2.2 | 2.3 |
| Whole grain, g/d | 12.0 | 17.2 | 26.4 | 16.7 | 22.4 | 31.4 |
| Whole grains: total grains in weight | 0.2 | 0.2 | 0.3 | 0.3 | 0.3 | 0.4 |
| Calcium supplement, % | 35 | 39 | 59 | 41 | 44 | 64 |

\* Updated information throughout follow-up was used to calculate the means for continuous variables and percentage for categorical variables. All variables are age- and sex-standardized except person-years, age, and sex.

† Healthy lifestyle score (range, 0–5) was defined as the number of the 5 healthy lifestyle factors: normal body weight (body mass index, ≥18.5 and <25.0 kg/m$^2$), never smoking or past smoking with pack-years <5, moderate-to-vigorous intensity activity for ≥30 minutes per day, none-to-moderate alcohol intake (<1 drink [14 g alcohol]/d for women and <2 drinks/d for men), and meeting at least 3 of the 6 dietary recommendations by the World Cancer Research Fund/American Institute for Cancer Research Third Expert Report 2018, which included red meat <0.5 serving/d, processed meat <0.2 serving/d, dietary fiber ≥30 g/d, dairy products ≥3 servings/d, whole grains ≥48 g/d or account for at least half of total grains, and calcium supplement use.

‡ Among ever smokers only.

screening, with the HR of 0.82 (95% CI, 0.74 to 0.91) and 0.89 (95% CI, 0.82 to 0.96), respectively (P-interaction = 0.27) (Fig 1B). Overall, endoscopic screening was associated with an HR for CRC incidence of 0.53 (95% CI, 0.49 to 0.58) and for CRC mortality of 0.49 (95% CI, 0.43 to 0.56).

When assessed by anatomical site and TNM stage, the associations remained similar between the screening and non-screening groups for proximal colon cancer (P-interaction = 0.29) and distal CRC (P-interaction = 0.93) (S2 Fig), and TNM 1 and 2 (P-interaction = 0.49) and TNM 3 and 4 CRC (P-interaction = 0.86) (S3 Fig). Similar associations were also found for each of the individual lifestyle factors between the screening and non-screening groups (all P-interaction > 0.10) (S1 and S2 Tables).

We then assessed the combined association of healthy lifestyle and endoscopic screening (Table 2). For CRC incidence, the PAR with endoscopic screening alone was 32% (95% CI, 31% to 33%), increased to 37% (95% CI, 33% to 41%) when combined with a healthy lifestyle score of ≥2, 43% (95% CI, 37% to 48%) with score ≥3, 49% (95% CI, 42% to 55%) with score ≥4, and 61% (95% CI, 42% to 75%) with score = 5. For CRC mortality, the PAR with

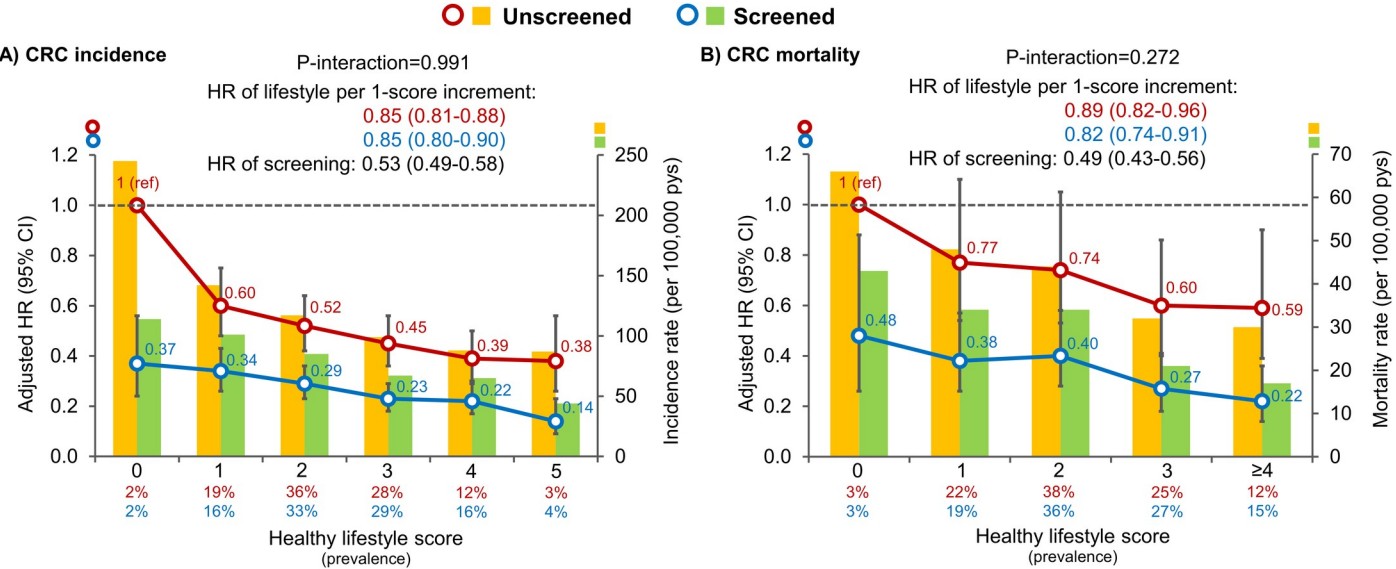

**Fig 1. Association of healthy lifestyle score with CRC incidence (panel A) and mortality (panel B) according to endoscopic screening status and the corresponding age-adjusted incidence and mortality rates.** Healthy lifestyle score (range, 0–5) was defined as the number of the 5 healthy lifestyle factors: normal body weight (BMI, ≥18.5 and <25.0 kg/m$^2$), never smoking or past smoking with pack-years <5, moderate-to-vigorous intensity activity for ≥30 minutes per day, none-to-moderate alcohol intake (<1 drink [14 g alcohol]/d for women and <2 drinks/d for men), and meeting at least 3 of the 6 dietary recommendations by the WCRF/AICR Third Expert Report 2018, which included red meat <0.5 serving/d, processed meat <0.2 serving/d, dietary fiber ≥30 g/d, dairy products ≥3 servings/d, whole grains ≥48 g/d or account for at least half of total grains, and calcium supplement use. Multivariable Cox proportional hazards regression was used to calculate the HRs and 95% CIs while adjusting for age, calendar period, sex, ethnicity, current multivitamins use, regular aspirin use, family history of CRC, menopausal status and hormone use (women only) in incidence analysis, and, additionally, diagnoses of cardiovascular disease and type 2 diabetes in mortality analysis. Error bars indicate 95% CIs. BMI, body mass index; CI, confidence interval; CRC, colorectal cancer; HR, hazard ratio; pys, person-years; WCRF/AICR, World Cancer Research Fund/American Institute for Cancer Research.

endoscopic screening alone was 34% (95% CI, 29% to 39%), increased to 38% (95% CI, 30% to 45%), 50% (95% CI, 40% to 59%), and 55% (95% CI, 37% to 69%) when combined with a lifestyle score of ≥2, ≥3, and ≥4, respectively.

We also calculated the PARs by CRC subsites (Fig 2) and TNM stage (Fig 3). For incidence of proximal colon cancer, the PAR increased from 15% (95% CI, 13% to 16%) with endoscopic screening alone to 51% (95% CI, 17% to 74%) with the combination of endoscopic screening and healthy lifestyle score of 5. The corresponding PAR for distal CRC increased from 47% (95% CI, 45% to 50%) to 75% (95% CI, 61% to 84%), for TNM 1 and 2 CRC from 27% (95% CI, 26% to 28%) to 52% (95% CI, 31% to 68%), and for TNM 3 and 4 CRC from 37% (95% CI, 33% to 41%) to 68% (95% CI, 45% to 84%). For mortality, combining a healthy lifestyle (score ≥4) with endoscopic screening was associated with an increase in the PAR from 13% (95% CI, 2% to 23%) to 52% (95% CI, 20% to 74%) for proximal colon cancer, and from 49% (95% CI, 42% to 56%) to 65% (95% CI, 48% to 78%) for distal CRC.

In the stratified analysis, we found that an integration of healthy lifestyle with endoscopic screening was associated with a substantially increased PAR across all the subgroups according to family history of CRC, regular aspirin use, age, and sex (S3 Table). In addition, we found a beneficial association of healthy lifestyle score with mortality due to other causes than CRC in both the endoscopic screening (HR, 0.86 per one-unit increment; 95% CI, 0.85 to 0.88) and non-screening groups (HR, 0.83 per one-unit increment; 95% CI, 0.82 to 0.84) (S4 Fig).

Our results were robust to several sensitivity analyses, including restricting to participants aged 50 to 75 years (S4 Table), defining screening based on colonoscopy only (S5 Table), stopping updating lifestyle information after the first report of endoscopic screening (S6 Table), excluding

**Table 2. PAR estimates for CRC incidence and mortality associated with endoscopic screening and healthy lifestyle\* separately and in combination.**

| Exposures | Prevalence (%) | Age-adjusted rate /100,000 pys | % PAR (95% CI)[†] |
|---|---|---|---|
| Colorectal cancer incidence | | | |
| Endoscopic screening | 41 | 78 | 32 (31–33) |
| Healthy lifestyle score ≥2 | 80 | 92 | 7 (5–10) |
| Healthy lifestyle score ≥3 | 45 | 82 | 16 (12–21) |
| Healthy lifestyle score ≥4 | 17 | 75 | 24 (16–31) |
| Healthy lifestyle score = 5 | 3 | 65 | 34 (16–49) |
| Endoscopic screening and healthy lifestyle score ≥2 | 34 | 73 | 37 (33–41) |
| Endoscopic screening and healthy lifestyle score ≥3 | 20 | 65 | 43 (37–48) |
| Endoscopic screening and healthy lifestyle score ≥4 | 8 | 61 | 49 (42–55) |
| Endoscopic screening and healthy lifestyle score = 5 | 2 | 44 | 61 (42–75) |
| Colorectal cancer mortality | | | |
| Endoscopic screening | 41 | 28 | 34 (29–39) |
| Healthy lifestyle score ≥2 | 76 | 33 | 5 (1–9) |
| Healthy lifestyle score ≥3 | 39 | 26 | 20 (11–28) |
| Healthy lifestyle score ≥4 | 13 | 24 | 23 (7–38) |
| Endoscopic screening and healthy lifestyle score ≥2 | 32 | 26 | 38 (30–45) |
| Endoscopic screening and healthy lifestyle score ≥3 | 17 | 20 | 50 (40–59) |
| Endoscopic screening and healthy lifestyle score ≥4 | 6 | 17 | 55 (37–69) |

CI, confidence interval; CRC, colorectal cancer; PAR, population-attributable risk; pys, person-years; WCRF/AICR, World Cancer Research Fund/American Institute for Cancer Research.

\* Healthy lifestyle score (range, 0–5) was defined as the number of the 5 healthy lifestyle factors: normal body weight (body mass index, ≥18.5 and <25.0 kg/m$^2$), never smoking or past smoking with pack-years <5, moderate-to-vigorous intensity activity for ≥30 minutes per day, none-to-moderate alcohol intake (<1 drink [14 g alcohol]/d for women and <2 drinks/d for men), and meeting at least 3 of the 6 dietary recommendations by the WCRF/AICR Third Expert Report 2018, which included red meat <0.5 serving/d, processed meat <0.2 serving/d, dietary fiber ≥30 g/d, dairy products ≥3 servings/d, whole grains ≥48 g/d or account for at least half of total grains, and calcium supplement use.

† PARs and 95% CIs were calculated while adjusting for age, calendar period, sex, ethnicity, current multivitamins use, regular aspirin use, family history of CRC, menopausal status and hormone use (women only) in incidence analysis, and, additionally, diagnoses of cardiovascular disease and type 2 diabetes in mortality analysis.

the person-times when response to endoscopy was missing (S7 Table), and defining the healthy lifestyle score using the 5 lifestyle factors as 5-categorical variables (S5 Fig and S8 Table).

## Discussion

In 2 large prospective cohorts, we found that adherence to a healthy lifestyle was associated with a similar risk reduction of CRC incidence and mortality among individuals with and without endoscopic screening. Our results suggest that approximately 32% of CRC cases and 34% of CRC deaths could potentially be prevented by endoscopic screening alone and, when combined with the 5 healthy lifestyle factors, these figures increased to 61% and 55%, respectively. Moreover, the incremental benefit was particularly strong for proximal colon cancer (PAR increased from 15% to 51% for incidence, and from 13% to 52% for mortality). These findings indicate that adherence to healthy lifestyle confers a substantial benefit for reduction of CRC incidence and mortality independent of endoscopic screening and that integrating healthy lifestyle with endoscopic screening may substantially enhance CRC prevention.

Screening is the cornerstone of CRC prevention, and endoscopic screening has been shown to confer a persisting protection against CRC for up to 15 to 17 years [32,33]. In the current study, we found that endoscopic screening was associated with an HR for CRC incidence of 0.53 and for CRC mortality of 0.49, consistent with our prior findings [33] and other studies

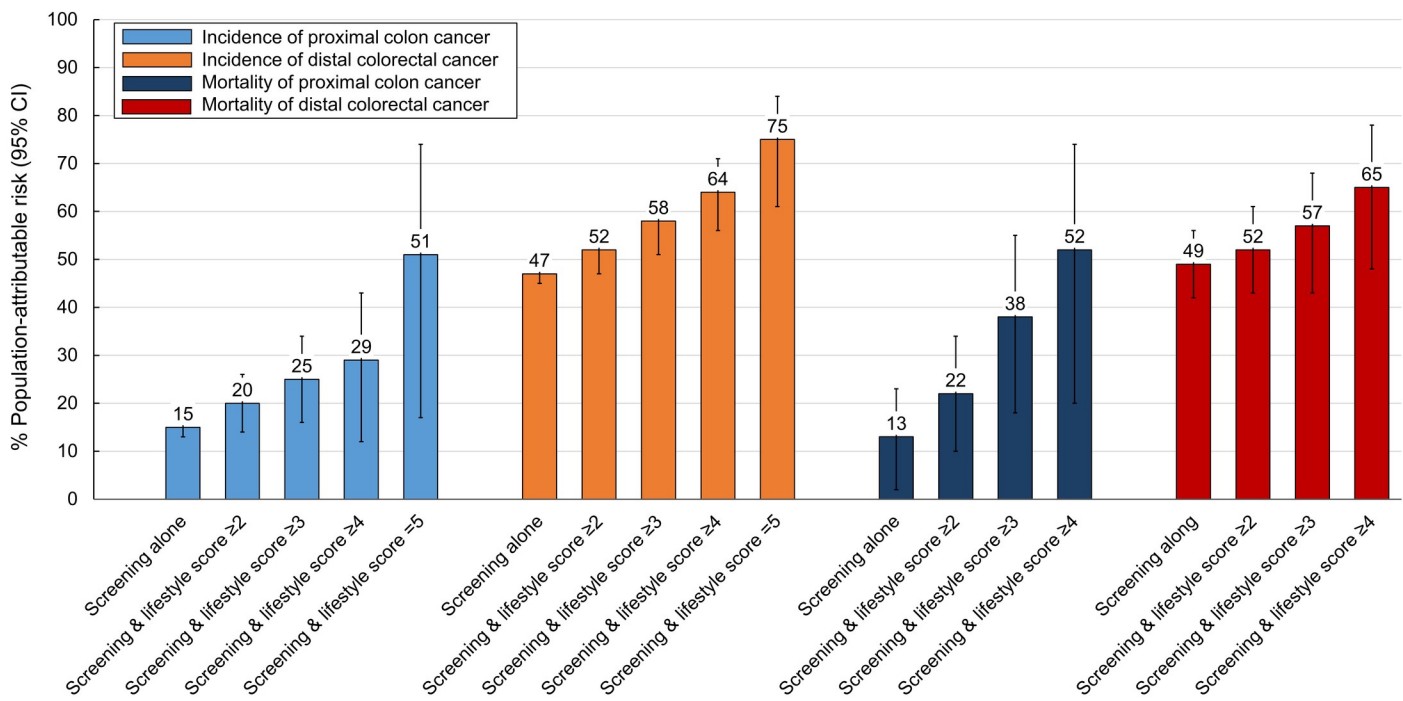

**Fig 2. PAR estimates for incidence and mortality of proximal colon cancer and distal CRC with endoscopic screening alone and endoscopic screening–healthy lifestyle combination.** Healthy lifestyle score (range, 0–5) was defined as the number of the 5 healthy lifestyle factors: normal body weight (BMI, $\geq$18.5 and <25.0 kg/m$^2$), never smoking or past smoking with pack-years <5, moderate-to-vigorous intensity activity for $\geq$30 minutes per day, none-to-moderate alcohol intake (<1 drink [14 g alcohol]/d for women and <2 drinks/d for men), and meeting at least 3 of the 6 dietary recommendations by the WCRF/AICR Third Expert Report 2018, which included red meat <0.5 serving/d, processed meat <0.2 serving/d, dietary fiber $\geq$30 g/d, dairy products $\geq$3 servings/d, whole grains $\geq$48 g/d or account for at least half of total grains, and calcium supplement use. PARs and 95% CIs were calculated while adjusting for age, calendar period, sex, ethnicity, current multivitamins use, regular aspirin use, family history of CRC, menopausal status and hormone use (women only) in incidence analysis, and, additionally, diagnoses of cardiovascular disease and type 2 diabetes in mortality analysis. Error bars indicate 95% CIs. BMI, body mass index; CI, confidence interval; CRC, colorectal cancer; PAR, population-attributable risk; WCRF/AICR, World Cancer Research Fund/American Institute for Cancer Research.

[34,35]. Over the years, despite an overall increase in the uptake of endoscopic screening, substantial disparity remains in the US [36,37]. On the other hand, although the importance of lifestyle and diet for CRC prevention has been increasingly recognized [12], it remains unclear whether adherence to healthy lifestyle in individuals having already undertaken endoscopic screening remains as beneficial for CRC prevention as in those without endoscopic screening. To the best of our knowledge, the current study represents the first effort to answer that question and to comprehensively quantify the joint beneficial association of healthy lifestyle and endoscopic screening with CRC prevention.

Although precancerous lesions can be detected and removed through endoscopic screening, the benefit is far from perfect, due to procedural (e.g., poor bowel preparation, difficulty in accessing the proximal colon) and lesion-related reasons (e.g., missed/incomplete resection of serrated polyps). On the other hand, lifestyle modification, as a primary prevention approach for CRC, operates through various biological pathways, including reduction in hyperinsulinemia and systemic inflammation, and modulation of gene expression and the gut microbiota [38–42]. Therefore, given the independent pathways through which healthy lifestyle and endoscopic screening exert protection against CRC, it is not surprising that we observed a similar CRC-preventive association of adherence to healthy lifestyle in individuals with and without endoscopic screening.

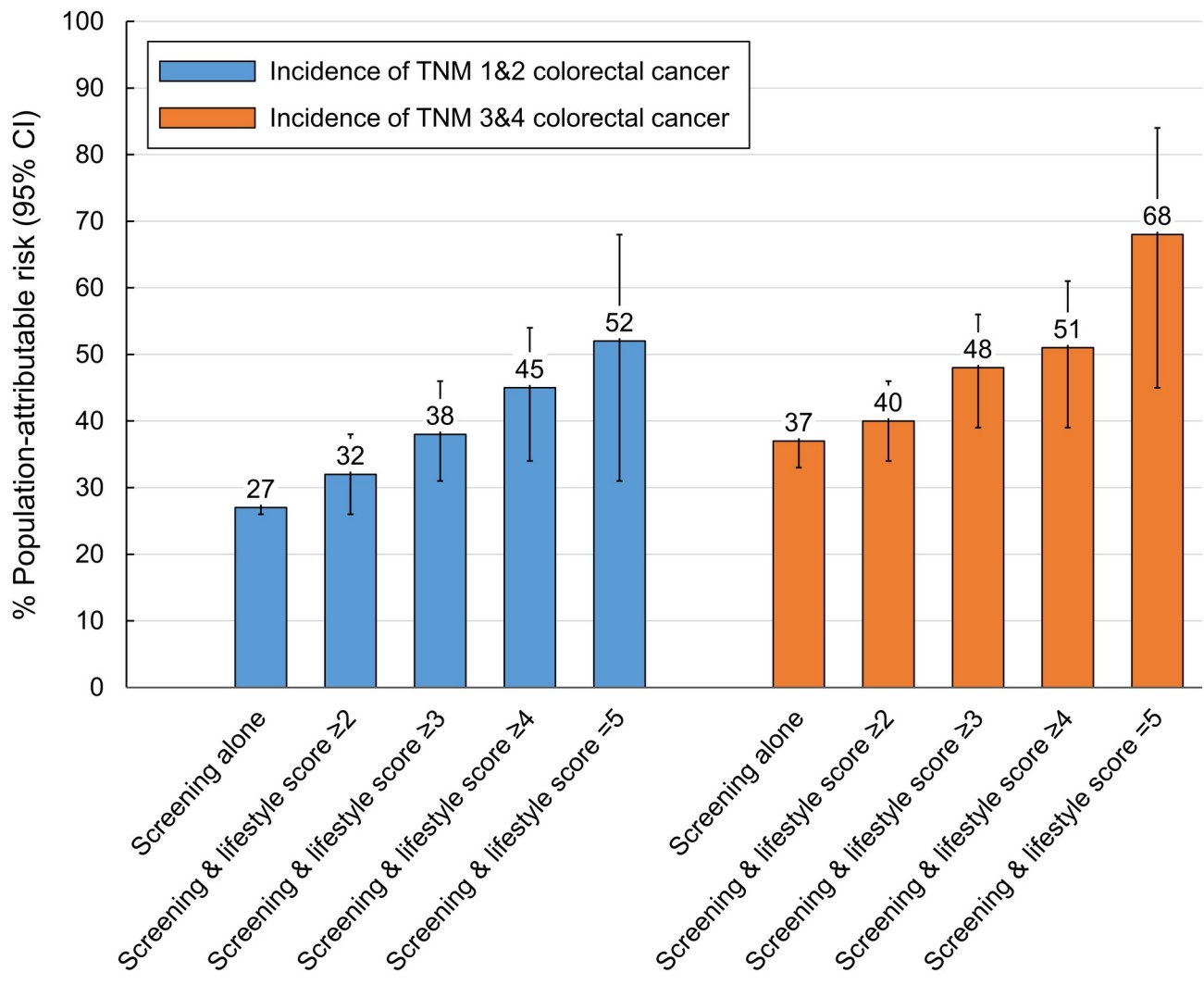

**Fig 3. PAR estimates for incidence of CRC by TNM stage (TNM 1 and 2 and 3 and 4) with endoscopic screening alone and endoscopic screening–healthy lifestyle combination.** Healthy lifestyle score (range, 0–5) was defined as the number of the 5 healthy lifestyle factors: normal body weight (BMI, $\geq$18.5 and <25.0 kg/m$^2$), never smoking or past smoking with pack-years <5, moderate-to-vigorous intensity activity for $\geq$30 minutes per day, none-to-moderate alcohol intake (<1 drink [14 g alcohol]/d for women and <2 drinks/d for men), and meeting at least 3 of the 6 dietary recommendations by the WCRF/AICR Third Expert Report 2018, which included red meat <0.5 serving/d, processed meat <0.2 serving/d, dietary fiber $\geq$30 g/d, dairy products $\geq$3 servings/d, whole grains $\geq$48 g/d or account for at least half of total grains, and calcium supplement use. PARs and 95% CIs were calculated while adjusting for age, calendar period, sex, ethnicity, current multivitamins use, regular aspirin use, family history of CRC, and menopausal status and hormone use (women only). Error bars indicate 95% CIs. BMI, body mass index; CI, confidence interval; CRC, colorectal cancer; PAR, population-attributable risk; TNM, tumor–node–metastasis; WCRF/AICR, World Cancer Research Fund/American Institute for Cancer Research.

Our findings have important clinical and public health implications. First, we provide empirical evidence for the independent associations of healthy lifestyle and endoscopic screening with lower CRC incidence and mortality. Despite the protection of endoscopic screening against CRC [32,33], the fact that healthy lifestyle may be associated with a further increased CRC prevention highlights the need to promote healthy lifestyle to optimize CRC prevention. Second, we observed a higher increment in the beneficial association for proximal colon cancer when integrating healthy lifestyle with endoscopic screening. Given the rising incidence of

proximal colon cancer and the diminished benefit of endoscopy in preventing these tumors [33], our results suggest that an emphasis on healthy lifestyle may help address these limitations. Third, our findings highlight the merit of healthy lifestyle for CRC prevention in settings lacking coverage or capacity for endoscopic screening [43,44]. For example, compared to individuals with no endoscopic screening or healthy lifestyle, those with the 5 healthy lifestyle practices but remain unscreened might have a 62% reduction in CRC risk (HR, 0.38; 95% CI, 0.26 to 0.56), which was very close to that (63%) among individuals only undertaking endoscopic screening but still living a less healthy lifestyle (HR, 0.37; 95% CI, 0.24 to 0.56). Fourth, we noted a more general benefit of healthy lifestyle than endoscopic screening for other outcomes than CRC. We found that each increment in the number of healthy lifestyle practices was associated with 14% to 17% reduction in mortality due to other causes than CRC. However, no such benefit was observed for endoscopic screening. Nevertheless, given the challenges in undertaking and maintaining healthy lifestyle practices (only 3% of our study population had all 5 healthy lifestyle factors), screening should still serve as an unshakable cornerstone for CRC prevention.

Our study has several strengths, including the large sample size, long-term follow-up, repeated assessments of lifestyle factors and endoscopic screening, and detailed information on potential confounders. Also, several limitations should be noted. First, the information of lifestyle factors and endoscopic screening was self-reported and thus subject to measurement error. However, the accuracy of these self-reported data within our cohorts has been well documented (see S2 Text) [22–24,33,45,46]. Second, we used a less stringent threshold to define healthy lifestyle due to the low prevalence of healthy lifestyle. For example, we included past smoking with pack-years <5 in the nonsmoking group and defined healthy lifestyle as a score of ≥4 rather than 5 in CRC mortality analysis. Thus, the PAR for healthy lifestyle may have been underestimated. Third, our study participants are all health professionals and predominantly whites, thereby limiting the generalizability of our findings, particularly given that individuals with a healthier lifestyle are more likely to undertake screening. However, it is unlikely that the biological effect of healthy lifestyle would substantially differ between health professionals and the general population. On the other hand, given the generally healthier lifestyle profiles of health professionals, we may have underestimated the benefit of lifestyle in the general population. Additionally, we lacked data on other screening modalities than endoscopy and were thus unable to examine their influence on our results.

In conclusion, our study suggests that adherence to healthy lifestyle is associated with substantial reduction in CRC incidence and mortality independent of endoscopic screening. Integration of healthy lifestyle with endoscopic screening may improve prevention of CRC, particularly proximal colon cancer, compared to endoscopic screening alone.

## Supporting information

**S1 Checklist. Strengthening the Reporting of Observational Studies in Epidemiology (STROBE) checklist.**
(DOCX)

**S1 Fig. Flowchart of study participants.**
(TIF)

**S2 Fig.** Association of healthy lifestyle score with incidence of proximal colon cancer (panel A) and distal colorectal cancer (panel B) according to endoscopic screening status and the corresponding age-adjusted incidence rates. Healthy lifestyle score (range, 0–5) was defined as the number of the 5 healthy lifestyle factors: normal body weight (BMI, ≥18.5 and <25.0 kg/m2),

never smoking or past smoking with pack-years <5, moderate-to-vigorous intensity activity for ≥30 minutes per day, none-to-moderate alcohol intake (<1 drink [14 g alcohol]/d for women and <2 drinks/d for men), and meeting at least 3 of the 6 dietary recommendations by the WCRF/AICR Third Expert Report 2018, which included red meat <0.5 serving/d, processed meat <0.2 serving/d, dietary fiber ≥30 g/d, dairy products ≥3 servings/d, whole grains ≥48 g/d or account for at least half of total grains, and calcium supplement use. Multivariable Cox proportional hazards regression was used to calculate the HRs and 95% CIs while adjusting for age, calendar period, sex, ethnicity, current multivitamins use, regular aspirin use, family history of CRC, and menopausal status and hormone use (women only). P-heterogeneity = 0.175 among unscreened participants and 0.349 among screened participants. BMI, body mass index; CI, confidence interval; CRC, colorectal cancer; HR, hazard ratio; pys, person-years; WCRF/AICR, World Cancer Research Fund/American Institute for Cancer Research.
(TIF)

**S3 Fig.** Association of healthy lifestyle score with incidence of CRC by TNM stage (panel A: TNM 1 and 2; panel B: TNM 3 and 4) according to endoscopic screening status. Healthy lifestyle score (range, 0–5) was defined as the number of the 5 healthy lifestyle factors: normal body weight (BMI, ≥18.5 and <25.0 kg/m2), never smoking or past smoking with pack-years <5, moderate-to-vigorous intensity activity for ≥30 minutes per day, none-to-moderate alcohol intake (<1 drink [14 g alcohol]/d for women and <2 drinks/d for men), and meeting at least 3 of the 6 dietary recommendations by the WCRF/AICR Third Expert Report 2018, which included red meat <0.5 serving/d, processed meat <0.2 serving/d, dietary fiber ≥30 g/d, dairy products ≥3 servings/d, whole grains ≥48 g/d or account for at least half of total grains, and calcium supplement use. Multivariable Cox proportional hazards regression was used to calculate the HRs and 95% CIs while adjusting for age, calendar period, sex, ethnicity, current multivitamins use, regular aspirin use, family history of CRC, and menopausal status and hormone use (women only). BMI, body mass index; CI, confidence interval; CRC, colorectal cancer; HR, hazard ratio; TNM, tumor–node–metastasis; WCRF/AICR, World Cancer Research Fund/American Institute for Cancer Research.
(TIF)

**S4 Fig. Association of healthy lifestyle score with mortality due to other causes than CRC according to endoscopic screening status.** Healthy lifestyle score (range, 0–5) was defined as the number of the 5 healthy lifestyle factors: normal body weight (BMI, ≥18.5 and <25.0 kg/m$^2$), never smoking or past smoking with pack-years <5, moderate-to-vigorous intensity activity for ≥30 minutes per day, none-to-moderate alcohol intake (<1 drink [14 g alcohol]/d for women and <2 drinks/d for men), and meeting at least 3 of the 6 dietary recommendations by the WCRF/AICR Third Expert Report 2018, which included red meat <0.5 serving/d, processed meat <0.2 serving/d, dietary fiber ≥30 g/d, dairy products ≥3 servings/d, whole grains ≥48 g/d or account for at least half of total grains, and calcium supplement use. Multivariable Cox proportional hazards regression was used to calculate the HRs and 95% CIs while adjusting for age, calendar period, sex, ethnicity, current multivitamins use, regular aspirin use, family history of CRC, menopausal status and hormone use (women only), diagnoses of cardiovascular disease and type 2 diabetes, physical exam for disease screening, mammography for breast cancer screening (women only), and prostate-specific antigen testing for prostate cancer screening (men only). BMI, body mass index; CI, confidence interval; CRC, colorectal cancer; HR, hazard ratio; WCRF/AICR, World Cancer Research Fund/American Institute for Cancer Research.
(TIF)

**S5 Fig. Association of healthy lifestyle score with CRC incidence according to endoscopic screening status when healthy lifestyle score was defined by the five 5-categorical lifestyle factors.** The five 5-categorical lifestyle factors included BMI (18.5–24.9, 25.0–27.4, 27.5–29.5, 30.0–34.9, and ≥35.0 kg/m$^2$; scored 5–1, respectively), smoking (never, past smoking with pack-years <5, past smoking with pack-years ≥5, current smoker with pack-years <20, and current smoker with pack-years ≥20; scored 5–1, respectively), physical activity (moderate-to-vigorous intensity activity for 0, 0.1–0.9, 1.0–3.4, 3.5–5.9, and ≥6 hours per week; scored 1–5, respectively), alcohol intake (0, 0.1–13.9, 14–20.9, 21–27.9, and ≥28 g/d; scored 5–1, respectively), and number of the 6 healthy dietary components recommendations by the WCRF/AICR Third Expert Report 2018 (0–1, 2, 3, 4, and 5–6; scored 1–5, respectively). The 6 healthy dietary components included red meat <0.5 serving/d, processed meat <0.2 serving/d, dietary fiber ≥30 g/d, dairy products ≥3 servings/d, whole grains ≥48 g/d or account for at least half of total grains, and calcium supplement use. Sum of the 5 scores (range, 5–25) was then categorical into 5 levels (5–10, 11–13, 14–16, 17–19, 20–22, and 23–25). These 5 levels were defined as the new healthy lifestyle score (0, 1, 2, 3, 4, and 5). Multivariable Cox proportional hazards regression was used to calculate the HRs and 95% CIs while adjusting for age, calendar period, sex, ethnicity, current multivitamins use, regular aspirin use, family history of CRC, and menopausal status and hormone use (women only). Error bars indicate 95% CIs. BMI, body mass index; CI, confidence interval; CRC, colorectal cancer; HR, hazard ratio; WCRF/AICR, World Cancer Research Fund/American Institute for Cancer Research.
(TIF)

**S1 Table. Associations of individual lifestyle factors with CRC incidence and mortality according to endoscopic screening status.**
(DOCX)

**S2 Table. Associations of individual lifestyle factors with incidence of proximal colon cancer and distal CRC according to endoscopic screening status.**
(DOCX)

**S3 Table. PAR estimates for CRC incidence and mortality with endoscopic screening alone and endoscopic screening–healthy lifestyle combination and the corresponding age-adjusted prevalence, stratified by family history of CRC, regular aspirin use, age, and sex.**
(DOCX)

**S4 Table. PAR estimates for CRC incidence and mortality with endoscopic screening and healthy lifestyle separately and in combination among the participants aged 50–75 years.**
(DOCX)

**S5 Table. PAR estimates for CRC incidence and mortality with colonoscopic screening and healthy lifestyle separately and in combination.**
(DOCX)

**S6 Table. PAR estimates for CRC incidence and mortality with endoscopic screening and healthy lifestyle separately and in combination when we stopped updating lifestyle information after the first endoscopic screening.**
(DOCX)

**S7 Table. PAR estimates for CRC incidence and mortality with endoscopic screening and healthy lifestyle separately and in combination when we excluded the person-time when**

**response to endoscopy was missing.**
(DOCX)

**S8 Table. PAR estimates for CRC incidence with endoscopic screening and healthy lifestyle separately and in combination when healthy lifestyle score was defined by the five 5-categorical lifestyle factors.**
(DOCX)

**S1 Text. Prospective analysis proposal.**
(DOCX)

**S2 Text. Detailed assessment and statistical analysis.**
(DOCX)

## Acknowledgments

The authors thank the participants and staff of the NHS and the HPFS for their continued contributions, as well as the following state cancer registries for their help: Alabama, Arizona, Arkansas, California, Colorado, Connecticut, Delaware, Florida, Georgia, Idaho, Illinois, Indiana, Iowa, Kentucky, Louisiana, Maine, Maryland, Massachusetts, Michigan, Nebraska, New Hampshire, New Jersey, New York, North Carolina, North Dakota, Ohio, Oklahoma, Oregon, Pennsylvania, Rhode Island, South Carolina, Tennessee, Texas, Virginia, Washington, and Wyoming. The authors assume full responsibility for analyses and interpretation of these data.

## Author Contributions

**Conceptualization:** Kai Wang, Mingyang Song.

**Data curation:** Kai Wang, Mingyang Song.

**Formal analysis:** Kai Wang, Wenjie Ma, Kana Wu, Andrew T. Chan, Edward L. Giovannucci, Mingyang Song.

**Funding acquisition:** Shuji Ogino, Andrew T. Chan, Edward L. Giovannucci, Mingyang Song.

**Investigation:** Kai Wang, Andrew T. Chan, Edward L. Giovannucci, Mingyang Song.

**Methodology:** Kai Wang, Wenjie Ma, Kana Wu, Shuji Ogino, Andrew T. Chan, Edward L. Giovannucci, Mingyang Song.

**Project administration:** Kai Wang, Wenjie Ma, Mingyang Song.

**Resources:** Kai Wang, Edward L. Giovannucci, Mingyang Song.

**Software:** Kai Wang, Wenjie Ma, Mingyang Song.

**Supervision:** Shuji Ogino, Andrew T. Chan, Edward L. Giovannucci, Mingyang Song.

**Validation:** Kai Wang, Mingyang Song.

**Visualization:** Kai Wang.

**Writing – original draft:** Kai Wang.

**Writing – review & editing:** Kai Wang, Wenjie Ma, Kana Wu, Shuji Ogino, Andrew T. Chan, Edward L. Giovannucci, Mingyang Song.

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
