## [Editor Report · Decision Letter 0]

4 Aug 2020

Dear Dr Song, 

Thank you for submitting your manuscript entitled "Healthy lifestyle, endoscopic screening, and colorectal cancer incidence and mortality in the US: a nationwide cohort study" for consideration by PLOS Medicine.

Your manuscript has now been evaluated by the PLOS Medicine editorial staff [as well as by an academic editor with relevant expertise] and I am writing to let you know that we would like to send your submission out for external peer review.

Kind regards,

Adya Misra, PhD,

Senior Editor

PLOS Medicine

---

## [Decision Letter · Decision Letter 1]

27 Aug 2020

Dear Dr. Song,

Thank you very much for submitting your manuscript "Healthy lifestyle, endoscopic screening, and colorectal cancer incidence and mortality in the US: a nationwide cohort study" (PMEDICINE-D-20-03426R1) for consideration at PLOS Medicine. 

[LINK]

In light of these reviews, I am afraid that we will not be able to accept the manuscript for publication in the journal in its current form, but we would like to consider a revised version that addresses the reviewers' and editors' comments. Obviously we cannot make any decision about publication until we have seen the revised manuscript and your response, and we plan to seek re-review by one or more of the reviewers. 

We expect to receive your revised manuscript by Sep 17 2020 11:59PM. Please email us (plosmedicine@plos.org) if you have any questions or concerns.

We look forward to receiving your revised manuscript. 

Sincerely,

Emma Veitch, PhD

PLOS Medicine

On behalf of Clare Stone, PhD, Acting Chief Editor,

PLOS Medicine

plosmedicine.org

*Please structure your abstract using the PLOS Medicine headings (Background, Methods and Findings, Conclusions - "methods and findings" is a single subsection).

*In the last sentence of the Abstract Methods and Findings section, please describe the main limitation(s) of the study's methodology.

*In the abstract conclusions, the text currently reads "Healthy lifestyle may reduce CRC incidence and mortality independent of

endoscopic screening", but this assumes that the effect sizes seen in the main analyses represent causal effects, and given the study design we can't assume this. It would be better in the abstract to use more moderate language and to present the findings consistently as associations - in the main discussion you can then go on to say that, if those effects were causal then the implications might mean (xyz...), but this conclusion shouldn't be leapt to.

*Please use the following style for the author summary - https://journals.plos.org/plosmedicine/s/revising-your-manuscript#loc-author-summary - this should use bullets and the headers need to be different. See also an example, https://journals.plos.org/plosmedicine/article?id=10.1371/journal.pmed.1002416

*We'd suggest ensuring that the study is reported according to the STROBE guideline; when doing so, please include the completed STROBE checklist as Supporting Information. Please add the following statement, or similar, to the Methods: "This study is reported as per the Strengthening the Reporting of Observational Studies in Epidemiology (STROBE) guideline (S1 Checklist)." The STROBE guideline can be found here: http://www.equator-network.org/reporting-guidelines/strobe/. When completing the checklist, please use section and paragraph numbers, rather than page numbers.

*Did your study have a prospective protocol or analysis plan? Please state this (either way) early in the Methods section.

*In the discussion section, as noted above for the abstract, some of the language should be moderated so it doesn't assume causal effects. Eg, "Approximately 32% of CRC cases and 34% of CRC deaths could potentially be prevented by endoscopic screening alone" - this could be presented later, but with the caveat that these estimates are based on assuming the effect size seen in your analyses are entirely reflective of a "true" causal effect (which may not be true). The discussion in general should be careful about where it uses causal language and it would be better to ensure the main part of the discussion presents the findings as associations, only going on later to discuss the implications **if these associations are indeed causal**. 

Comments from the reviewers:

Reviewer #1: I confine my remarks to statistical aspects of this paper. The general approach is OK but I have a number of issues to resolve before I can recommend publication.

First, quoting a "per unit" estimate is misleading, as the trend in both groups was non-linear (and nonlinear in different ways).

Second, the interaction term added is not adequate, as it does not address the above. Consider CRC incidence (as shown in fig 1A). If the factor of 0.85 was correct, then the nonscreened group would have HRs of 1, 0.85, 0.72, 0.61, 0.52 and 0.44; while the screened group would have values of 0.37 0.31, 0.27 0.23 0.19 amd 0.16. These are not accurate (and the two lines are not parallel)

Third, it's an interesting question whether the appropriate measure for reduction in risk is the ratio or the difference. Reducing a large risk by (say) 15% is not the same as reducing a small risk by 15%. 

More specific points: On p. 5 at the top, give a rate of CRC death per person, not just "3rd highest"

Also on p. 5, the huge range of risk reduction for liefestyle (20 to 70%) makes these claims less credible (at least to me). 

p. 7 Please define "metabolic equivalents"

p. 7 I'm not altogether happy with the way the lifestyle factor was computed. Dichotomoizing risk factors is kind of silly. People don't go from "normal" to "high" risk when they cross some threshold. E.g., for BMI (a very flawed measure of obesity) you aren't fine at 24.9 and then high risk at 25.1. Also, 31 minutes of exercise is not dramatically different from 29 minutes. It would be better to develop a measure of risk that left all measures continuouis (and maybe nonllinear).eith based on data or based on literature. 

p 10 How was the "per unit" number calculated, given that there were 5 different logistic regressions run for each group?

Peter Flom

Reviewer #2: Wang and colleagues address an important topic- the associations between screening and/or healthy lifestyles and the risk of colorectal cancer incidence and mortality, using the Nurses health Study and Health professionals follow-up study, two large, important cohorts that have been sources for many important observational studies. They found that the protective effects of screening and health lifestyle (based on a healthy lifestyle score) were independently associated with lower risks of CRC incidence and mortality. These findings are important and timely. The study is well-executed and the supplementary analyses helpful. I would make a few suggestions to better put them in context:

1) I would ask that the language be changed to highlight these are associations, not interventions or trials: For example, instead of concluding: "Healthy lifestyle may reduce CRC incidence and mortality independent of endoscopic screening." I would suggest "Healthy lifestyle is associated with lower CRC incidence and mortality independent of endoscopic screening." Similarly, saying "First, we provide empirical evidence for the substantial benefit of lifestyle modification, independent of endoscopic screening, for improved CRC prevention." is probably less accurate than "First, we provide empirical evidence for the independent association of healthy behaviors and endoscopic screening with lower CRC incidence."

2) It would be helpful to discuss the rationale for turning the actual, sometimes linear measures included in the healthy lifestyle score into dichotomized scores. Giving the same weight to 3 drinks per day and 10 drinks per day, for example, seems to underweight extreme scores.

3) What is the relationship between the lifestyle score and all-cause mortality? I am assuming the the effects go beyond that of CRC.

4) The discussion talks about "lifestyle modification" which is an intervention framing, but I think it is important to reiterate the associations are with given levels of healthy behaviors moreso than the results of chnages in those behaviors. This framing is important because while both screening and healthy lifestyles are associated with less incidence and mortality, the effort required to achieve the protective levels of lifestyle (medium to high intensity counseling interventions) may be considerably greater than those required to get endoscopic screening (a discrete event).

Reviewer #3: From non-randomized studies on colonoscopy screening as well as from randomized studies on sigmoidoscopy and FOBT studies we have learned that endoscopic screening is a sufficient tool to reduce incidence and mortality from colorectal cancer. Furthermore, epidemiologigal studies have identified various life style associated risk factors for colorectal cancer, including BMI, smoking, nutrition. Probably up to 50% of colorectal cancers could be prevented by a healthy life style. Whereas both a healthy life style and screening can reduce the colorectal cancer burden and both are recommended by various national and international guidelines, it is so far not known whether individuals undergoing screening colonoscopy can further reduce their colorectal cancer risk by maintaining a healthy life style.

Wang et al. provide an answer to this important question in their study. Based on the analysis of well-known and thoroughly studied cohorts, the nurses health study and the health professionals follow-up study, they are able to demonstrate that the reduction of colorectal cancer incidence and mortality by maintaining a healthy life style is independent of participating in colonoscopy screening. Furthermore, individuals who undergo colonoscopy and follow a healthy life style are able to further reduce their colon cancer risk. Therefore, the recommendation will be to participate in bowel screening and to maintain a healthy lifestyle in oder to minimize risk. To the best of my knowledge, this ist he first study to provide this evidence.

The study is well designed, the analyzes have been properly performed and the manuscipt is well written. I only have some minor issues:

1. Is there data available on the adenomatous polyp count / size in the colonoscopies? If yes, is there a difference regarding life style? I would expect that a healthy life style is associated with a lower polyp count.

2. Were the colonoscopies primary screening colonoscopies? Or did some patients have sigmoidoscopies or FOB testing before with a finding that resulted in referral to secondary screening by colonoscopy? How did you include negative screening by sigmoidoscopy or negative FOBT screening in your analysis? Somebody having negative sigmoidoscopy and/or negative FOBT every other year has undergone screening that will also reduce the risk of a diagnosis of colorectal cancer and will usually not undergo colonoscopy. If data on FOBT screening is available, data for sigmoidoscopies is obviously available, it would be important to study the group with no screening at all as the unscreened group currently comprises no screeing at all + screening other than colonoscopy.

3. How did authors account for changes in life style, e.g. unhealthy lifestyle in the beginning of the study and changing to a healthier diet over time?

4. Was there a difference in Duke`s or UICC staging of the cancers diagnosed between groups? As a stage shift towards lower stages by screening is known, it could be speculated that the group with screening + healthy diet not only has less cancers but also cancers at earlier stages.

5. Was there a difference in risk of a colon cancer diagnosis between individuals receiving one or more negative screening colonoscopies over time? 

6. Page 8: Participants were considered as endoscopically unscreened until the first time they reported undertaking endoscopy for screening purpose and as endoscopically screened thereafter… How did authors account for diagnostic colonoscopies performed for complaints that did not provide any pathologies? A person undergoing colonoscopy for other reason than screening with no significant findings will usually not undergo screening colonoscopy within the next couple of years.

[LINK]

---

## [Decision Letter · Decision Letter 2]

23 Nov 2020

Dear Dr. Song,

Thank you very much for submitting your manuscript "Healthy lifestyle, endoscopic screening, and colorectal cancer incidence and mortality in the US: a nationwide cohort study" (PMEDICINE-D-20-03426R2) for consideration at PLOS Medicine. 

[LINK]

In light of these reviews, I am afraid that we will not be able to accept the manuscript for publication in the journal in its current form, but we would like to consider a revised version that addresses the reviewers' and editors' comments. Obviously we cannot make any decision about publication until we have seen the revised manuscript and your response, and we plan to seek re-review by one or more of the reviewers. 

We expect to receive your revised manuscript by Dec 14 2020 11:59PM. Please email us (plosmedicine@plos.org) if you have any questions or concerns.

We look forward to receiving your revised manuscript. 

Sincerely,

Adya Misra, PhD

Senior Editor 

PLOS Medicine

plosmedicine.org

Please compare the spline fit and linear fit as suggested by Ref1 in order to check what the differences are and if they are of practical importance rather than statistical significance. Please look at the reduction in error between the two models and then judge whether it is big enough to make the more complex spline model worth the effort.

Please add "our results indicate" or "our study suggests" in order to tone down the language throughout. 

Please avoid implying a causal relationship between healthy lifestyle, endoscopic screening and colorectal cancer as this is an observational study 

Please introduce all acronyms on first view. For instance the study cohorts are mentioned in the methods but it is not clear what the "NHS" stands for, for example. 

Please add participant demographics in the abstract 

Comments from the reviewers:

Reviewer #1: The authors have addressed most of my concerns. The remaining issue is about the splines.

I checked "proceed without recommendation" because what to do depends on non-statistical issues.

Here is the statistics part:

The authors tested a cubic spline for statistical significance and found it was not significant. But the p value of a test for linearity is not really the point. P values, in general, are problematic (see the American Statistical Ass'n Statement https://medium.com/@peterflom/some-thoughts-on-9-11-4f746ae9bdcb) and, here, as often, the real issue is effect size. That is, are the curves shown in the figure in their response to me straight or not?

The same effect size will be significant with more subjects. The key issue is whether it is meaningful and important.

And that's a non-statistical question. It depends on whether the difference between a straight line estimate and a spline estimate is "worth it" in terms of complexity. 

One way to help answer this question is to see how well the two models do - how much does the spline model reduce the error? Is it enough that people in the field will care? 

Peter Flom

Reviewer #3: The revised version of the manuscript has been strongly improved. My questions have been sufficiently answered.

[LINK]

---

## [Decision Letter · Decision Letter 3]

8 Dec 2020

Dear Dr. Song,

Thank you very much for re-submitting your manuscript "Healthy lifestyle, endoscopic screening, and colorectal cancer incidence and mortality in the US: a nationwide cohort study" (PMEDICINE-D-20-03426R3) for review by PLOS Medicine.

I have discussed the paper with my colleagues and the academic editor and it was also seen again by xxx reviewers. I am pleased to say that provided the remaining editorial and production issues are dealt with we are planning to accept the paper for publication in the journal.

[LINK]

We look forward to receiving the revised manuscript by Dec 15 2020 11:59PM.   

Sincerely,

Adya Misra, PhD

Senior Editor 

PLOS Medicine

plosmedicine.org

Requests from Editors:

Please revise to "United States" in the title

Please capitalise "W" in whites

Page 6- paragraph 2 suggest revising “high quality diet” as it is currently ambiguous

Page 7 suggest removing “prospectively”

Please add citations to or copies of the questionnaires used in both NHS and HPFS cohorts

Page 12 Results 1st paragraph should say “multivitamins”

Page 16- please remove "remarkable"

Discussion-I suggest removing all iterations of “causal” as this is an observational study. In addition, I suggest removing the speculation about biological pathways linking healthy lifestyle and CRC incidence.

Late in the paper you note that colonoscopy can include removal of CRC precursors. Please mention this as a possible basis for the preventive effect in the author summary and/or introduction. I assume there are no data on this.

Please remove funding and conflicts of interest information from the main text. These should be provided in the relevant sections of the article submission form.

Presumably, those less likely to be screened are less likely to have a healthy lifestyle - please mention this as a possible limitation?

Discussion- please avoid labels like “unhealthy lifestyle” I suggest replacing with less healthy lifestyle or similar

STROBE checklist- some information appears to be missing, for example page 3

Comments from Reviewers:

Reviewer #1: The authors have addressed my concerns and I now recommend publication

Peter Flom

[LINK]

---

## [Editor Report · Decision Letter 4]

15 Dec 2020

Dear Dr Song, 

On behalf of my colleagues and the Academic Editor, Dr Kolligs, I am pleased to inform you that we have agreed to publish your manuscript "Healthy lifestyle, endoscopic screening, and colorectal cancer incidence and mortality in the United States: a nationwide cohort study" (PMEDICINE-D-20-03426R4) in PLOS Medicine.

PRESS

Sincerely, 

Richard Turner, PhD 

rturner@plos.org